# Innovating Strategies and Tailored Approaches in Neuro-Oncology

**DOI:** 10.3390/cancers14051124

**Published:** 2022-02-22

**Authors:** Alberto Picca, David Guyon, Orazio Santo Santonocito, Capucine Baldini, Ahmed Idbaih, Alexandre Carpentier, Antonio Giuseppe Naccarato, Mario Caccese, Giuseppe Lombardi, Anna Luisa Di Stefano

**Affiliations:** 1Institut du Cerveau-Paris Brain Institute-ICM, Sorbonne Université, Inserm, CNRS, AP-HP, Hôpital Universitaire La Pitié Salpêtrière, DMU Neurosciences, 75013 Paris, France; picca.alberto@gmail.com (A.P.); ahmed.idbaih@aphp.fr (A.I.); 2Department of Medical Oncology, Gustave Roussy University Hospital, 94800 Villejuif, France; david.guyon@gustaveroussy.fr; 3Division of Neurosurgery, Spedali Riuniti di Livorno—USL Toscana Nord-Ovest, 57124 Livorno, Italy; orazio.santonocito@uslnordovest.toscana.it; 4Drug Development Department (DITEP), Gustave Roussy University Hospital, 94800 Villejuif, France; capucine.baldini@gustaveroussy.fr; 5Service de Neurochirurgie, Hôpital Universitaire La Pitié Salpêtrière, 75013 Paris, France; alexandre.carpentier@aphp.fr; 6Department of Translational Research and New Technologies in Medicine and Surgery, Division of Pathology, University of Pisa, 56100 Pisa, Italy; giuseppe.naccarato@unipi.it; 7Anatomia Patologica 1, Department of Laboratory Medicine, Pisa University Hospital, 56126 Pisa, Italy; 8Department of Oncology, Oncology 1, Veneto Institute of Oncology IOV-IRCCS, 35128 Padua, Italy; mario.caccese@iov.veneto.it (M.C.); giuseppe.lombardi@iov.veneto.it (G.L.); 9Department of Neurology, Foch Hospital, 92150 Suresnes, France

**Keywords:** glioma, molecular markers, targeted therapies, immunotherapy, blood–brain barrier disruption

## Abstract

**Simple Summary:**

Diffuse gliomas, including the most aggressive subtype glioblastoma, represent the most frequent primary central nervous system tumors. Despite intense chemoradiation protocols that represent the current standard of care, these cancers inevitably recur, and median overall survival does not exceed 18 months. New therapeutic options are compellingly needed for these tumors, particularly those lacking the favorable prognostic marker *IDH* mutation. Nonetheless, potentially druggable alterations are increasingly identified in distinct subsets of patients harboring gliomas. Targeted treatments, along with improved immunotherapeutic schedules, gene therapy, cell therapy, and physical strategies to improve drug delivery to the nervous system, are currently under extensive investigation. They bring hope for more effective therapies in these diseases with currently often a dismal outcome.

**Abstract:**

Diffuse gliomas, the most frequent and aggressive primary central nervous system neoplasms, currently lack effective curative treatments, particularly for cases lacking the favorable prognostic marker *IDH* mutation. Nonetheless, advances in molecular biology allowed to identify several druggable alterations in a subset of *IDH* wild-type gliomas, such as *NTRK* and *FGFR-TACC* fusions, and *BRAF* hotspot mutations. Multi-tyrosine kinase inhibitors, such as regorafenib, also showed efficacy in the setting of recurrent glioblastoma. IDH inhibitors are currently in the advanced phase of clinical evaluation for patients with *IDH*-mutant gliomas. Several immunotherapeutic approaches, such as tumor vaccines or checkpoint inhibitors, failed to improve patients’ outcomes. Even so, they may be still beneficial in a subset of them. New methods, such as using pulsed ultrasound to disrupt the blood–brain barrier, gene therapy, and oncolytic virotherapy, are well tolerated and may be included in the therapeutic armamentarium soon.

## 1. Introduction

Gliomas represent approximately 26–40% of primary central nervous system (CNS) tumors, with estimated incidence rates of around 6 cases per 100,000 population/year (more than 22,600 newly diagnosed patients in the USA and 26,600 in the European Union every year) [1,2,3]. Glioblastoma (GBM), a grade 4 glioma [4], accounts for 60–70% of all malignant gliomas. The incidence of GBM increases with age and in males [1]. Glioblastoma has a dismal prognosis, with a median overall survival despite standard chemoradiation of 15 months, a 2-year survival rate of around 27% and a 5-year survival rate of only 9.8% [5].

Management of glioma patients is severely impacted by the absence of effective curative treatments, the limited number of therapeutic options, and the intrinsic clinical and biological heterogeneity even within the same histological subgroup. Recent advances in molecular biology allowed to refine the diagnostic and prognostic classification of gliomas and are paving the way for a personalized medicine targeting the main driver oncogenes at a patient level [6,7,8,9,10,11].

In this review, we will discuss the most relevant diagnostic molecular markers of diffuse gliomas and their role in improving the classification of CNS tumors, the main prognostic markers, and eventually the “theranostic” markers with their corresponding targeted therapies currently under study for glioma patients. We will also present innovative and promising strategies from recent clinical trials.

## 2. The Integrated Histo-Molecular Classification and Personalized Management of Adult Diffuse Gliomas

During recent decades, brain tumor classification has been primarily based on the histological concept that these neoplasms could be classified according to their microscopic similarities with the hypothesized cells of origin and the presumed level of differentiation [12]. Nonetheless, increased knowledge demonstrated that several acquired molecular alterations allow a better definition of the different biological entities and their clinical aggressiveness [8,9,13,14].

Based on these advancements, since 2016, the World Health Organization (WHO) introduced the new paradigm of “integrated” histo-molecular classification [15]. In the WHO classification of adult diffuse gliomas, the status of two molecular alterations–namely, hotspot mutations in the genes coding for the isocitrate dehydrogenase [IDH] isoforms 1 and 2, and the chromosomal codeletion 1p/19q, an unbalanced translocation resulting in the complete loss of the 1p and 19q chromosomal arms, is crucial in glioma taxonomy, irrespective of the histological grading. Accordingly, adult diffuse gliomas have been separated into three broad groups [4,15].

Astrocytomas are defined by the presence of an *IDH* mutation without 1p/19q codeletion. This group includes less aggressive grade 2 and 3 gliomas, but also gliomas with histological grade 4 features (i.e., necrosis and/or microvascular proliferation), corresponding to the malignant progression of a former lower grade glioma and previously indicated as “secondary GBMs”. For them, the new definition of “astrocytoma, grade 4” has been established [4,16]. The presence of the homozygous deletion of the *CDKN2A* gene defines a group of patients with a worse prognosis [14], and it allows alone the classification of a tumor as “astrocytoma, grade 4” regardless of the histological features [4].

Oligodendrogliomas are specifically defined by the presence of the 1p/19q codeletion, that invariably associate with an *IDH* mutation. They constitute a subgroup of grade 2–3 gliomas with the best prognosis and a pronounced chemo- and radio-sensitivity.

*IDH* wild-type gliomas are the most aggressive entity regardless of histological grading [17]. The presence of at least one of the following: (i) *EGFR* gene amplification; (ii) chromosome 7 gain plus chromosome 10 loss; (iii) hotspot *TERT* promoter mutation, considered molecular markers of GBM, is sufficient to define an *IDH* wild-type diffuse glioma as GBM, independently of its histological appearance [4,18], although this remains partly disputed [17].

The identification of these three histo-molecular entities enables to personalize the treatment strategy (Table 1) [19,20].

Grade 4 gliomas (both *IDH* mutated astrocytomas and *IDH* wildtype GBMs): patients with a good Karnofsky performance status (KPS > 70) and age less than 65 years are treated with six weeks concomitant chemoradiation followed by adjuvant temozolomide (the so-called “Stupp protocol”) [5,21]. For patients older than 65 years and/or a KPS < 70, 3-week hypofractionated radiotherapy is recommended [22], with or without temozolomide. The O6-methylguanine-DNA-methyltransferase (*MGMT*) promoter methylation status may be helpful in clinical decision making of adding alkylating temozolomide, although this remains debated (*see above*).

Grade 3 oligodendroglioma (*IDH* mutated and 1p/19q co-deleted) patients receive radiotherapy plus the procarbazine, CCNU, and vincristine (PCV) polychemotherapy [25,26].

Grade 3 astrocytomas (*IDH* mutated and 1p/19q non co-deleted) are treated with radiotherapy plus adjuvant temozolomide [27,29] or adjuvant PCV [25].

Grade 2 gliomas, *IDH* mutated (with or without the 1p/19q codeletion) considered at high risk because of the age at diagnosis (more than 40 years) or the presence of a residual tumor after surgery: the benefit of the radiotherapy plus the PCV chemotherapy has been reported in a phase 3 randomized clinical trial [28].

Grade 2 and 3 gliomas, *IDH* wild-type: they are currently considered aggressive tumors and patients are often treated with the Stupp protocol [17].

## 3. The Clinical Utility of the *MGMT* Gene Promoter Methylation Status

*MGMT* gene promoter (p*MGMT*) methylation is a well-recognized predictive marker of sensitivity to alkylating agents in *IDH* wildtype gliomas [30,31,32,33]. *MGMT* gene codes for a DNA repair enzyme that removes mutagenic alkyls from the O6 position of guanine. The promoter hypermethylation results in reduced gene expression (epigenetic silencing). Patients with p*MGMT* hypermethylation have an increased progression-free survival and a 21.7 months overall survival after standard chemoradiation, significantly longer than those without gene promoter hypermethylation [30].

Nonetheless, the clinical benefit at the 5-years timepoint of the concomitant treatment with radiochemotherapy compared to radiotherapy alone seems to be also present in a part of the p*MGMT* non-methylated patient cohort [5]. Consequently, the p*MGMT* methylation status should not be considered as a formal discriminant while choosing the therapeutic strategy to be adopted in the first line setting of a young, otherwise healthy patient with a newly diagnosed GBM.

On the other hand, in elderly or frail subjects, who could be ineligible to radiotherapy, the presence of p*MGMT* hypermethylation and its predictive value for response to alkylating agents could motivate the initiation of treatment with chemotherapy alone rather than upfront palliative care, due to the reasonable hope of functional and neurological improvement [23,24,34]. Conversely, in elderly patients with unmethylated p*MGMT*, the benefit/risk balance favors radiotherapy alone [19].

Given its major predictive value, p*MGMT* methylation status is now considered crucial in the design of clinical trials of patients with newly diagnosed GBM [35,36].

## 4. Theranostic Markers and Targeted Treatments

In the recurring setting, therapeutic options are limited. Nitrosoureas [37] and/or the antiangiogenic agent bevacizumab [38] are usually discussed as second line treatments [19,20]. Concerning the potential targeted therapies, we will discuss in the following paragraphs the most promising actionable pathways in patients affected by gliomas.

## 5. Tyrosine Kinase Inhibition

### 5.1. Multi-Kinase Inhibitors

Tyrosine kinase receptors are transmembrane proteins involved in several cellular processes, including cell differentiation, regulation of proliferation, survival, metabolism, cell cycle control and cell migration. Because of their well-recognized oncogenic potential, they have been largely studied in oncology and several targeted compounds capable of blocking their activity have been developed. Furthermore, in neuro-oncology, different lines of research have explored the ability of different tyrosine kinase inhibitors (TKIs) to improve patient outcome, especially in the recurrent setting. Encouraging data derive from the use of regorafenib (Figure 1A–D), an oral multi-kinase inhibitor targeting VEGFR1–3, PDGFR, TIE2, FGFR, RAF-1, KIT, RET, and BRAF. The drug has been evaluated for the treatment of recurrent GBM patients in the randomized phase II trial REGOMA (Table 2) [39]. In this study, the use of regorafenib resulted in a significant benefit in terms of 6 months progression free and overall survival, as well as in terms of disease control rate compared to the standard of care lomustine, with a manageable toxicity profile [39]. Subsequent studies evaluated the presence of any predictors of response to regorafenib in patients with recurrent GBM. The activation of the AMPK pathway appeared associated with a clinical benefit from treatment with regorafenib in the same patient population [40], while the expression of 2 gene transcripts (*HIF1A*, *CDKN1A*) and 3 miRNAs (miR-3607-3p, miR-301a-3p, miR-93-5p) could help identify a subgroup of GBM patients exhibiting a striking survival advantage when treated with regorafenib [41]. Based on these data, the NCCN guidelines have included regorafenib as the preferred regimen in cases of recurrent GBM. The combination of regorafenib and the anti PD-1 immune-checkpoint inhibitor (ICI) nivolumab is currently evaluated in the ongoing phase II multi-indication study (NCT04704154, Table 2), but results are not yet available.

Another response adaptive randomization platform phase II/III trial is currently active, with the aim of evaluating multiple treatment regimens for newly diagnosed and recurrent GBM patients (GBM AGILE, NCT03970447). A Bayesian response adaptive randomization allocates the enrolled patients in different treatment arms including regorafenib, temozolomide, lomustine, the PI3K/AKT/mTOR inhibitor paxalisib, and the alkylating agent VAL-083 (Table 2).

### 5.2. MAP-Kinase Pathway Inhibition

The proto-oncogene *BRAF* codes for the B-Raf serine/threonine kinase, part of the Raf kinase protein family involved in the activation of the oncogenic Mitogen-Activated Protein Kinase (MAPK) pathway. The *BRAF* V600E mutation is a recurrent alteration in xanthoastrocytomas, glioneuronal tumors, pilocytic astrocytomas, and, less frequently, diffuse astrocytomas [49,50]. The total frequency of *BRAF* mutations in gliomas remains below 6%.

This alteration is involved in tumoral proliferation; the therapeutic opportunity stems from the actionable nature of the *BRAF* V600E mutation, initially recognized in non-neurological tumors, notably metastatic melanoma [51]. BRAF V600E inhibitors (BRAFi) reduce MAPK phosphorylation thereby affecting apoptosis and inhibiting the progression in the cell cycle. The association with MAPK/ERK Kinase (MEK) inhibitors (MEKi), acting downstream in the same pathway, increase the signal blockade and improve clinical safety [52,53]. 

Clinical responses may vary from prolonged responses with a remarkable clinical benefit to primary resistance to the targeted therapy. The response rate in *BRAF* V600E-mutant gliomas exceeds 30%, with associated clinical benefit and prolonged tumor control [42] (Figure 1E–H). Rechallenging with BRAFi +/− MEKi at recurrence in patients initially responding to these targeted therapies may also be effective, as recently reported [54].

It should be noted that targeted approaches with BRAF inhibitors could provide fruitful options in other *BRAF* V600E mutated brain tumors such as the aggressive or rapidly progressive papillary craniopharyngioma [55] and rhabdoid meningioma meningioma [50]. Although the estimated prevalence of *BRAF* V600 mutations in GBM is low (about 2%) [49,50], given its significant therapeutic implication, routine screening for *BRAF* mutations should also be strongly encouraged in this setting.

### 5.3. Inhibition of FGFR3-TACC3 Gene Fusions and Activating Mutations of FGFR1 Gene

The disruption of the Fibroblast Growth Factor Receptors (FGFR) pathway is a recurrent alteration affecting approximately 7% of solid cancers. Involved mechanisms may be ligand-dependent or independent, such as gene amplifications, activating mutations, and chromosomal translocations, which all lead to aberrant activation of the tyrosine kinase domain [56]. Several small molecule inhibitors, ligand traps, and monoclonal antibodies are currently being tested in various cancers [56].

Two rare FGFR alterations have been consistently reported in gliomas: fusions involving the families of genes *FGFR* and *TACC* (mostly *FGFR3-TACC3*) [57,58], and hotspot mutations N546 and K656 in *FGFR1* gene [59], showing a major therapeutic convergence from the possibility of treating patients with targeted anti-FGFR compounds. The inhibition of FGFR kinase resulted in clinical benefit in patients harboring the oncogenic fusion *FGFR3-TACC3* [60] (Figure 2), and different FGFR inhibitors are currently on trial (AZD4547, NCT02824133; TAS120, NCT02052778; erdafitinib, NCT04083976) (Table 2).

Hotspots *FGFR1* N546 and K656 mutations are recurrent in adult midline gliomas (affecting the thalamus, diencephalon, brainstem, and spine) [61,62], with a reported incidence of up to 18% regardless of grading, location, histological type, and other molecular alterations [59]. Based on the activating effect of these mutations [61], patients harboring these alterations have been treated with FGFR inhibitors [63] (TAS120, NCT02052778) (Table 2), results are currently awaited.

Screening for *FGFR3-TACC3* fusions and *FGFR1* activating mutations should be performed in all patients with newly diagnosed *IDH* wild-type gliomas and all midline gliomas, respectively, as these patients are potentially eligible for clinical trials of targeted therapies at recurrence (Table 2).

### 5.4. NTRK Pathway Inhibition

Three different genes (*NTRK1*, *NTRK2*, and *NTRK3*) encode for the Neurotrophic-Tropomyosin Receptor tyrosine Kinases (TRK). Fusions involving the *NTRK* genes are actionable oncogenic drivers involved in several cancers [64]. Rapid and sustained responses have been obtained in 60 to 80% of cases, including advanced or metastatic disease, with TRK inhibitors. An efficacy superior to 50% has been observed in paediatric gliomas [65].

The incidence of *NTRK* gene fusions in adult glioma patients remains rare, estimated at 2% [66]. The use of TRK small molecule inhibitors (first generation: larotrectinib and entrectinib, second generation: selitrectinib and repotrectinib) has shown to induce dramatic, durable responses in patients with primary or metastatic brain lesions [43,67,68] (Figure 1I–L), demonstrating blood–brain barrier crossing and intracranial activity. Phase I and II basket trials are currently ongoing in solid tumors including brain tumors [66] (Table 2).

Nonetheless, resistance phenomena may be observed. A frequent escape mechanism is the occurrence of a new mutation in the receptor tyrosine kinase. Resistance by activation of the MAPK pathway may also occur; hence the interest, as in the case of *BRAF* mutations, of associating an anti-MEK treatment [69].

### 5.5. Other Tyrosine Kinase Inhibitors

*EGFR* gene alterations are present in approximately 25% of gliomas [7]. Several EGFR TKI have been evaluated as possible treatments in patients diagnosed with glioma. Erlotinib and gefitinib, first-generation EGFR inhibitors, despite having shown interesting data in the preclinical setting [70], did not then lead to an improvement in the outcome and response parameters in the treatment of patients with GBM, both as first-line treatment and at relapse [71,72,73,74,75]. Second-generation inhibitors (afatinib and dacomitinib), again, did not result in a clinical benefit for GBM patients, showing limited activity both in combination with temozolomide and as single-agents [76,77].

Other small molecules, VEGFR TKI, have been evaluated as possible treatments in glioma patients: cediranib [78], sorafenib [79,80], sunitinib [81], pazopanib [82], and cabozantinib [83] were tested without however significant results in terms of responses and outcomes [84]. 

A further target of interest is PDGFR, another tyrosine kinase receptor often overexpressed in high-grade gliomas [85]. Imatinib, a TKI capable of blocking PDGFR, although it has shown evident efficacy in various types of cancer, has not shown significant activity in high-grade gliomas, neither alone nor in association with hydroxyurea [86,87]. Tandutinib, an oral PDGFRβ kinase inhibitor that demonstrated activity in patients with relapsed and refractory acute myelocytic leukemia, was also tested in patients with relapsed high-grade glioma in a phase II study in combination with bevacizumab. It showed an efficacy comparable to that of bevacizumab alone, with neuro-muscular junction pathologies as distinctive toxicity [88].

## 6. Additional Approaches Targeting EGFR Alterations

The Epidermal Growth Factor Receptor (EGFR) is part of the broad group of receptor tyrosine kinases; according to The Cancer Genome Atlas (TCGA) data, alterations of *EGFR* gene are present in around 25% of gliomas: 54% of GBMs and 9% of lower grade gliomas [7]. Unfortunately, several small molecule TKI approved for systemic cancers have shown disappointing results in the setting of high-grade glioma patients (*see above*).

Regarding *EGFR* gene amplification, the phase II trial testing ABT-414 (also known as depatuxizumab mafodotin), an anti-EGFR monoclonal antibody conjugated to a potent antimitotic agent (monomethyl auristatin F), in patients with a recurring GBM with EGFR amplification, observed a response in 39% of patients [26]. Unfortunately, the phase III trial including patients with newly diagnosed GBM (INTELLANCE1, NCT02573324) [89] has been interrupted for futility.

The EGFR truncated transcript variant III, or EGFRvIII, is a molecular alteration that results in constitutive pathway activation and is found in approximately 20% of GBMs [7]. Rindopepimut is a peptide vaccine directed against EGFRvIII. The compound has been tested in a phase III randomized clinical trial (ACT IV, NCT01480479), that has been regrettably interrupted because of futility [90]. The main hypotheses to explain this failure have been, on the one hand, the heterogeneous expression of EGFRvIII, that, by selection pressure, results in the proliferation of tumor cells without the targeted alteration, and, on the other hand, the instability of the EGFRvIII antigen during the course of the disease [91]. Nonetheless, the concomitant approach of peptide vaccination combined with immunostimulating compounds (namely, ICIs, *see below*) remains a promising field requiring further clinical research.

## 7. IDH Inhibition

*IDH* gene mutation is an early event in gliomagenesis [92,93] and plays a crucial role in initiating and sustaining astrocytomas and oligodendrogliomas growth. *IDH* mutated gliomas represent a distinct molecular entity among gliomas, in terms of evolution, prognosis, and response to treatments [94,95]. *IDH* hotspot mutations result in a neomorphic IDH enzymatic activity with the consequent accumulation of the oncometabolite D-2-hydroxyglutarate (D2HG) [96]. D2HG inhibits several α-ketoglutarate depending enzymes, including the *TET* family of DNA demethylases [97]. This results in a specific epigenetic signature, corresponding to a diffuse genome hypermethylation (glioma CpG island methylator phenotype, G-CIMP) [98] and consequent cellular dedifferentiation sustaining tumor growth. Several approaches are currently in study to target the enzymatic and epigenetic peculiarities of *IDH* mutated gliomas [94,95].

Several small molecule inhibitors of IDH have been developed [95] (Table 2). Ivosidenib (AG-120) is an oral inhibitor of the IDH1 mutated enzyme, while enasidenib (AG-221) inhibits mutated IDH2; both are approved for the treatment of acute myeloid leukaemia (AML). A phase I basket trial exploring the feasibility of ivosidenib treatment in *IDH* mutated solid tumors (NCT02073994) included 66 patients with advanced gliomas. The drug was well tolerated, and showed signs of activity, particularly in non-enhancing diffuse gliomas (66.7% of them showed a reduction of tumor volume, with a median reduction of 6-months tumor volume growth rate from 26% pretreatment to 9% under treatment) [46]. 

Vorasidenib (AG-881) is a dual inhibitor of both mutant IDH1 and IDH2. Phase I trial NCT02481154 included 52 patients with *IDH* mutated gliomas recurring after or not responders to standard treatment. Again, non-enhancing tumors showed relevant rates of response (18% objective response rate, with a tumor volume reduction as the best response in 17/22 patients), while no objective responses were seen in enhancing tumor patients [47].

Based on these results, and other data suggesting that the *IDH* mutation may not be necessary for tumor maintenance in advanced phases of glioma progression [99,100,101], the use of IDH inhibitors is currently tested in earlier disease stages. INDIGO trial (NCT04164901, currently recruiting) is a phase 3 study that will evaluate vorasidenib in the setting of residual or recurrent, non-enhancing grade 2 *IDH*1/2 mutated gliomas after surgery only. 

Other approaches are evaluating the effects of the *IDH* mutation and accumulation of D2HG on tumor immune microenvironment, and how to exploit them therapeutically. *IDH* mutated tumors show less tumor infiltrating lymphocytes (TIL) compared to the *IDH* wildtype counterpart [102], and escape to natural killer cells via the epigenetic silencing of the NKG2D ligands [103]. Furthermore, D2HG produced by tumor cells may act as a paracrine signal that inhibits TIL activity [104]. The inhibition of IDH mutated enzyme activity could thus reverse, at least partially, local immunosuppression. A phase II study of the ivosidenib–nivolumab association in advanced *IDH*1 mutated solid tumors, including contrast-enhancing gliomas, is recruiting (NCT04056910) (Table 2).

Demethylating agents, such as azacytidine or decitabine, may reverse the hypermethylator phenotype and promote cells differentiation [94,95]. A phase II trial with subcutaneous administration of azacytidine (AGIR; NCT03666559) is ongoing in recurrent grade II and III *IDH* mutated gliomas (Table 2). Results are awaited.

## 8. Immunotherapy

High grade gliomas are known to induce an immunosuppressive microenvironment, with a low, dysfunctional lymphocytic infiltration and a prevalence of immunosuppressive, protumoral myeloid cells, in an immune-privileged environment, such as the CNS [105,106]. Conspicuous efforts have been made to therapeutically reverse this “cold” immune phenotype, although with modest results to date. Preclinical studies of ICIs showed a promising signal of oncological activity in gliomas [107]. Nonetheless, recent clinical trials failed to show proof of efficacy. The use of the anti-programmed cell death protein 1 (PD1) nivolumab failed to show a survival benefit compared to bevacizumab in recurrent GBM in phase III trial Checkmate-143 (ref. [108]). Similarly, the two phase III trials (Checkmate-498 and 548) investigating nivolumab in the first-line setting (*MGMT*-unmethylated and *MGMT*-methylated GBM, respectively) did not show benefits in terms of progression-free and overall survival [35,36]. Several mechanisms have been hypothesized to account for glioma resistance to ICI treatment [105,109]. These include, among others, a profound, irreversible T cell dysfunction, with the upregulation of alternative immune checkpoints (such as TIM3, TIGIT, LAG3) at baseline or as escape mechanisms to ICIs. Furthermore, glioma patients are often in a state of systemic immunosuppression, due to conventional treatments (such as glucocorticoids, radiation therapy, and temozolomide) [110,111], but also to the effects of the tumor itself [112]. The immune-privileged intracranial location could also be of relevance, although clinical benefit has been obtained from ICI in the treatment of brain metastases [113]. Ongoing trials are evaluating different strategies to improve ICI results in glioma patients. A phase I evaluating the feasibility of the anti-LAG3 relatlimab (BMS-986016) with or without nivolumab in recurrent GBM completed accrual, results are awaited (NCT02658981). Indoleamine 2,3-dioxygenase 1 (IDO) is an enzyme produced by GBM TME implicated in the impairment of cytotoxic T lymphocytes functions and upregulation of Tregs [114]. Phase I-II trials are ongoing to evaluate the combination of different IDO inhibitors and anti-PD1 compounds (NCT03532295, NCT04047706) (Table 2). The timing of ICI administration is also under active evaluation, as some results suggest a clinical benefit from the neoadjuvant administration of anti-PD1 pembrolizumab [115].

Tumors with microsatellite instability (MSI) exhibit a large number of somatic mutations (whereas GBM has typically a low mutation rate [116]) and are thought to be more sensitive to ICI treatment via the increased transcription of tumoral neoantigens. The success of pembrolizumab in cancers with microsatellite instability paved the way for its investigation in gliomas with MSI. This was one of the first successes of a basket trial, targeting tumors with MSI regardless of the histological type [117], that resulted in tissue-agnostic FDA approval of pembrolizumab in MSI-high cancers. Pembrolizumab has been subsequently FDA-approved for tissue-agnostic treatment of tumors with a high mutational burden (TMB-H, conventionally considered as >10 mutations per megabase) irrespectively of microsatellite status, based on the positive results of KEYNOTE-158 trial [118]. Nonetheless, the benefit of ICI treatment in TMB-H gliomas is less clear [119,120]. Gliomas rarely (e.g., <2% of cases) present a TMB-H (or “hypermutated”) phenotype at diagnosis (“de novo” hypermutated), but often acquire a hypermutated status at recurrence after standard temozolomide treatment (“acquired” hypermutation) [119], mostly through defective mismatch repair (MMR) system [119,121]. Interestingly, a recent study suggests that TMB-H cancers most likely to respond to ICI treatments are those with a relevant CD8+ lymphocytic infiltration [122]. Hypermutated gliomas were among those with a reduced CD8+ infiltration. Indeed, two recent studies, albeit with limited sample sizes, did not show a clear clinical benefit of ICI treatment in (mostly acquired) hypermutated gliomas [119,123]. De novo hypermutated glioma patients may experience an increased benefit from ICI [124], but further evidence is required. Clinical trials are ongoing to prospectively evaluate the role of ICI treatment in hypermutated gliomas (e.g., NCT04145115, Table 2).

## 9. CAR T Cells Therapy

Chimeric antigen receptor (CAR) T cells (CART) are engineered T lymphocytes expressing synthetic receptors (CARs) that allow the recognition of specific molecules and T cell activation in an HLA-unrestricted manner [125]. First-generation CARs are composed of an extracellular domain responsible for antigen recognition, a transmembrane domain, and an intracellular domain including the CD3ζ chain transducing T cell activation. Further modifications resulted from the addition of one (second-generation CARs) or more (third-generation CARs) intracellular costimulatory domains such as CD28 or 4-1BB. Fourth-generation CARs, after recognition of the target antigen, can induce the engineered lymphocyte to express proinflammatory cytokines, bi-specific T cell engagers (BiTEs), or other genes of interest [126,127].

The use of CART targeting tumor-expressed antigens (as CD19) led to remarkable results in hematologic malignancies [125]. Conversely, however, to date, clinical trials exploring CART in GBM patients led to only anecdotally benefits [128] while in most cases they did not show evidence of relevant antitumor effects [129,130,131,132] and arose not-negligible safety issues [131]. Therapeutic targets tested in initial phase I/II trials included interleukin-13 receptor subunit alpha-2 (IL-13Rα2) [128,129], EGFRvIII [130,131], and human epidermal growth factor receptor 2 (HER2) [132], chosen because of the selective expression in tumor cells compared to healthy brain cells. In the cited clinical trials, IL-13Rα2-directed CART was infused into the surgical cavity after resection of recurrent GBM, whereas EGFRvIII- and HER2-directed CART were administered intravenously. Little or no clinically relevant benefit was seen in most cases, despite analysis of re-resected tumors confirmed in situ trafficking of intravenously infused EGFRvIII-directed CAR T-cells [130]. A notable clinical response has been reported after intraventricular infusion of IL-13Rα2-directed CART in a patient with multifocal craniospinal recurrent GBM [128]. The observed benefit lasted 7.5 months, but the tumor eventually relapsed [128]. 

Several mechanisms have been evoked for these disappointing results. Firstly, tumor cells may escape CART losing the expression of the targeted molecule. Indeed, antigen loss has been demonstrated in tumors treated with IL-13Rα2- and EGFRvIII-directed CART [129,130]. It is unclear if this has been induced by administered treatments or representing the natural evolution of the disease, as EGFRvIII loss may occur even in the absence of EGFRvIII-directed treatments [91]. The intrinsic intratumor heterogeneity of GBM under therapeutical pressure could lead also to the selection of subclones lacking the expression of the molecule of interest. Furthermore, tumor specimens obtained from CART-treated patients display increased expression of coinhibitory molecules and increased Treg infiltration [130]. Finally, it should not be underscored that CART therapy is not devoid of potentially detrimental side effects [133], including neurological toxicity [134]. In the NCT01454596 trial, two patients developed respiratory failure shortly after intravenous administration of EGFRvIII-directed CART, and one died [131].

New approaches to overcome the discussed shortcomings include the utilization of new-gen CART, and the exploration of novel targets [126]. In this regard, the Brown group identified chlorotoxin (CLTX), a peptide derived from scorpion venom, as a promising tumor-binding peptide to be incorporated in CAR as the antigen-recognizing domain [135]. CLTX was demonstrated to bind the majority of tumor cells in more than 90% of tested tumor samples, with little to no reactivity with healthy brain and independently from the expression of other targets as IL-13Rα2, EGFRvIII, and HER2 (ref. [135]). CLTX-CART demonstrated promising activity in in vitro and murine glioma models [135] and is currently under investigation in a phase I trial (NCT04214392) (Table 2). Nonetheless, the same authors demonstrated that the expression of matrix-metalloproteinase 2 (MMP2) is required for an efficient tumor targeting of CLTX-CART [135]. Such as, it has been anticipated that loss of MMP2 in GBM cells could represent a tumor escape mechanism [127]. Results are awaited. Other strategies aim at targeting multiple tumor-associated antigens in order to overcome GBM heterogeneity. Ahmed and colleagues developed “universal” CAR (UCAR) co-targeting HER2, IL13Rα2, and ephrin A receptor 2 (EphA2), with promising preclinical results [136] but no clinical data available to date. Finally, a GBM-directed synNotch CART has been recently developed [137]. SynNotch receptors are engineered transmembrane receptors that after the recognition of the target antigen activates the expression of a specific transcript. Choe et al. recently reported engineered T lymphocytes that can conditionally express EphA2 and IL13Rα2-directed CAR under the control of a synNotch receptor recognizing both a tumor-specific but heterogeneous antigen (EGFRvIII) or an organ-specific antigen (myelin oligodendrocyte glycoprotein, or MOG) [137]. In patient-derived tumor xenograft, both EGFRvIII and MOG-directed synNotch-CART displayed higher antitumor effects, with reduced exhaustion, and no evidence of off-tumor killing [137].

## 10. Gene Therapy and Virotherapy Approaches

### 10.1. Gene Therapy

The term “gene therapy” indicates the administration of exogenous genetic material as a therapeutical intervention [138,139,140]. In cancer, it can be used to reactivate the expression of lost tumor suppressor genes, inhibit aberrantly activated oncogenes, or induce the expression of immunostimulatory molecules or suicide enzymes, the latter converting a non-toxic compound into a cytotoxic molecule [138,139,140]. Vectors that can deliver the genetic material to target cells include viruses (adenoviruses, retroviruses, lentiviruses), non-polymeric, and polymeric nanoparticles (as liposomes) [138]. A widely studied gene therapy approach in gliomas is the induction of the expression of suicide genes. One example is the herpes simplex virus thymidine kinase (HSV-TK) enzyme that renders cells sensible to ganciclovir prodrug [139]. A first phase III study using a retroviral vector in newly diagnosed HGG failed to show an increased survival in patients transduced with HSV-TK [141]; treatment failure has been associated with inefficient gene transduction. Subsequent phase II–III trials assessing adenoviral vectors [142,143,144] disclosed more favorable results and suggested that patients that receive a gross total resection are more likely to benefit from the treatment [143]. Another suicide gene codes for the cytosine deaminase (CDA) enzyme that can convert systemically delivered non-toxic 5-fluorocytosine into cytotoxic compound 5-fluorouracil [139]. An extensively explored vector is Toca-511 retrovirus, that is characterized by a retained replicating capacity. Other than causing the expression of CDA in target cells, Toca-511 seems able to partially reverse the local immunosuppression in gliomas [145]. Despite promising results in the setting of recurrent high grade gliomas in the phase II trial [146], a subsequent phase III study failed to show a survival benefit in transduced cases compared to the standard of care [147]. A further strategy relies on the local induction of expression of proinflammatory, antitumor cytokines, such as interferon β or γ or interleukin-12 (IL12). IL12, in particular, is a potent anticancer cytokine, but its systemic use is severely limited by relevant side effects. The local administration of an adenoviral vector (Ad-RTS-hIL-12) that induces the expression of IL12 under the control of an oral activator (veledimex) was acceptably safe and showed signs of efficacy in a recent phase I trial [148]. Re-resected tumors displayed increased CD8+ lymphocytes, mostly expressing PD1. This provided the rationale for a combinatorial approach testing the association of the Ad-RTS-hIL-12 gene therapy with the anti-PD1 nivolumab [149]. In the phase I trial, the combination displayed no additional toxicity compared to IL12 gene monotherapy [149], and a phase II trial (NCT04006119, Table 2) has been completed, results are awaited. Gene therapy can be used also to target protumoral cells in the tumor microenvironment. A remarkable example is VB-111, that uses an adenoviral vector to transduce a chimeric gene coding for a protein composed of the extracellular domain of the tumor necrosis factor (TNF) receptor 1 and an intracellular derived from Fas protein [139]. Under TNF stimulation, the chimeric protein elicits cellular apoptosis. Its expression can be limited to proliferating endothelial cells using a modified preproendothelin promoter [139]. Of note, differing from most gene therapy vectors targeting tumor cells that are administered locally in the surgical cavity, VB-111 is administered intravenously. Treatment with VB-111 showed to be safe in a phase I/II trial that enrolled recurrent GBM patients [150]. Furthermore, patients in the primed combination group (receiving VB-111 monotherapy at inclusion and switched to bevacizumab plus VB-111 continuation at tumor progression) displayed promising median overall survival (414 days) and 12-month survival rate (57%) [150]. Nonetheless, a subsequent phase III trial testing the combination of VB-111 plus bevacizumab failed to showed a survival benefit compared to bevacizumab alone [151]. It has been hypothesized that the lack of VB-111 monotherapy priming could explain for the absence of survival benefit seen in the primed combination subgroup in phase I/II trial [151]. A new phase II randomized, placebo-controlled trial (NCT04406272, Table 2) is currently ongoing to evaluate the utility of VB-111 in the neoadjuvant setting of recurrent glioblastoma (presurgical versus postsurgical administration).

### 10.2. Oncolytic Viruses

Oncolytic viruses are engineered or naturally oncoselective viruses that infect, replicate, and lyse tumor cells releasing new progeny capable to infect neighbor cells [138,140]. By infecting and lysing cells, they can also stimulate local immune responses. One of the most studied oncolytic viruses is human herpes virus type 1 (HSV1). A seminal work by Martuza and colleagues demonstrated that attenuated HSV1 can retain oncolytic activity while being unable to replicate in non-dividing cells as healthy neurons [152]. Further engineered HSV1 as HSV1716 (with deletion of the γ34.5 gene) and G207 (with deletion of the U_L_39 gene) display increased oncolytic selectivity [138,140]. Phase I trials in recurrent glioma patients demonstrated a good safety profile and feasibility, with evidence of viral replication in tumor cells [153,154,155,156]. Another oncolytic virus clinically tested in glioma patients is DNX-2401, a modified human adenovirus that selectively infects cells with impaired retinoblastoma pathway (a common oncogenic alteration seen in glioblastoma [7]). A recent phase I trial of intratumoral administration of DNX-2401 for recurrent malignant glioma demonstrated promising results in terms of sustained clinical responses (five out of 25 patients survived more than three years, with a >95% tumor response in three) [157]. A phase II combination trial of DNX-2401 plus pembrolizumab (KEYNOTE-192 CAPTIVE, NCT02798406) complete accrual. The first results, presented in abstract form, are promising in terms of safety and antitumoral activity [158,159], and a phase III trial is awaited.

### 10.3. Combinatory Approaches

It has been postulated that virotherapy efficacy may be limited by incomplete transduction of the target mass in greater lesions [160]. To counteract this shortcoming, combination therapies have been developed that bring together the conditional cytotoxic and immune-stimulatory approaches [138,160]. The concomitant administration of an adenoviral vector expressing HSV-TK (Ad-TK) that renders transduced cells sensitive to ganciclovir and an adenoviral vector expressing Flt3L (Ad-Flt3L), a small molecule crucial for dendritic cells (DC) development, has been shown to have synergistic effects [161]. Preclinical models demonstrated that, after Ad-TK + Ad-Flt3L treatment, DC are activated by damage-associated molecular pattern molecules as HMGB1 and after uptake of tumor antigens from dying cells they prime a systemic immune response [162]. Tumor responses can be further increased combining Ad-TK + Ad-Flt3L treatment with DC vaccination [163] or ICI [164]. A phase I clinical trial of Ad-TK + Ad-Flt3L treatment for newly diagnosed malignant glioma (NCT01811992) has been completed. Interim analyses showed an acceptable safety profile, with evidence of increased inflammatory infiltrate in re-resected tumors [165].

## 11. Blood–Brain Barrier Disruption by Pulsed Ultrasound

The intratumoral delivery of systemically administered therapies may be limited by the present because of the intracranial location of gliomas, particularly for infiltrative regions where the blood–brain barrier (BBB) is mostly intact. The intracranial drug delivery may be increased by disrupting the BBB using new physical methods such as low-intensity pulsed ultrasound (LIPU) in combination with systemic administration of micron-sized bubbles [166]. LIPU has proven to disrupt the BBB [167] (Figure 3) and increase the intracerebral concentrations of systemically administered antitumoral compounds [168].

LIPU has furthermore been shown to enhance survival in preclinical glioma models [170] and to be safe in long-term studies in nonhuman primates [171]. A recent phase 1 study evaluated the safety and feasibility of an intracranial ultrasound device (SonoCloud-1) used to disrupt the BBB and increase carboplatin delivery [48]. BBB disruption was visible on post-treatment T1-weighted MRI scans for most of the sonications performed. Repeated sonications in some patients resulted in tumor reduction in the field of the implant [48]. Treatment-related adverse events (transient cerebral oedema) were transient and manageable, without carboplatin-related neurotoxicity. Patients with BBB disruption clearly visible on MRI had increased PFS and OS compared with patients without evidence of BBB disruption [48]. The phase I/II study Sonocloud-9 treating recurrent GBM patients with LIPU and carboplatine (NCT03744026) completed accrual, and results are awaited. Phase II SonoFIRST study (NCT04614493) is currently enrolling newly diagnosed GBM patients to be treated with LIPU plus concurrent chemoradiation and adjuvant temozolomide.

## 12. Conclusions

The efficacy of conventional antineoplastic treatments remains very limited in patients with malignant gliomas. Many targeted approaches validated in general oncology have been tested also in neuro-oncology, with variable results. Activating molecular alterations of current therapeutic relevance remain infrequent but may greatly impact patients’ management when present. Similarly, while revolutionizing the therapeutic scenario in several advanced cancers, immunotherapies did not show a clear clinical benefit in the diffuse glioma setting. Several new approaches to increase their efficacy are currently under investigation. Hypermutated gliomas (both de novo and treatment-induced) may be most likely to benefit from ICI treatments, although this remains a matter of discussion. These observations encourage to systematically screen for druggable alterations and a hypermutated status since the initial diagnosis and to discuss surgery at recurrence in order to obtain new specimens to be tested for. New methods, such as pulsed ultrasound to disrupt the blood–brain barrier, gene therapy, and oncolytic virotherapy, are well tolerated and may be included in the therapeutic armamentarium soon.

## Figures and Tables

**Figure 1 cancers-14-01124-f001:**
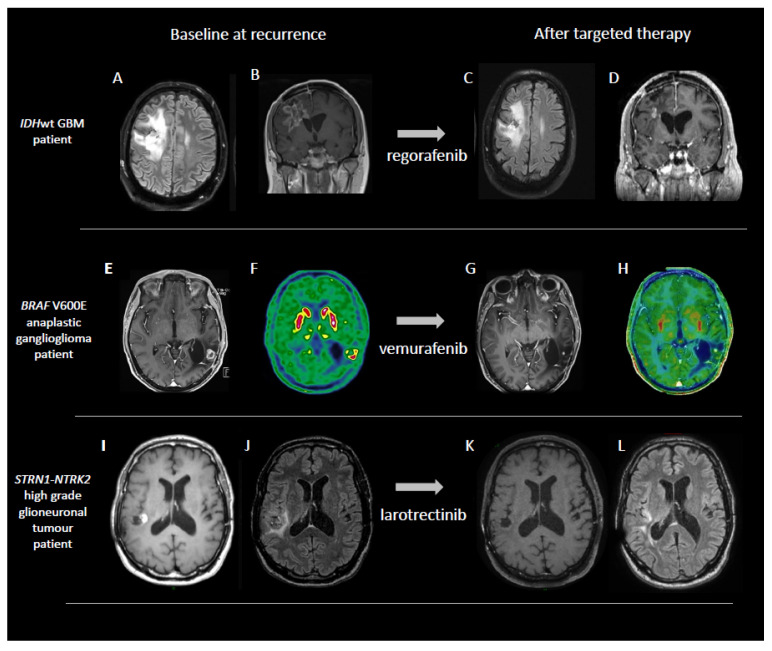
Examples of objective responses to tyrosine kinase inhibition in patients with primary brain tumors. **Panels** (**A**–**D**): tumor response after two cycles of regorafenib in a 49-year-old patient with recurrent *IDH* wild-type GBM. **Panels** (**E**–**H**): tumor response after three cycles of vemurafenib in a 38-year-old patient affected by recurrent *BRAF* mutant anaplastic ganglioglioma (case already reported in ref. [42]). **Panels** (**I**–**L**): a 53-year-old patient with *STRN1-NTRK2* fusion positive high grade glioneuronal tumor treated with larotrectinib and experiencing a complete tumor response (case already reported in ref. [43]).

**Figure 2 cancers-14-01124-f002:**
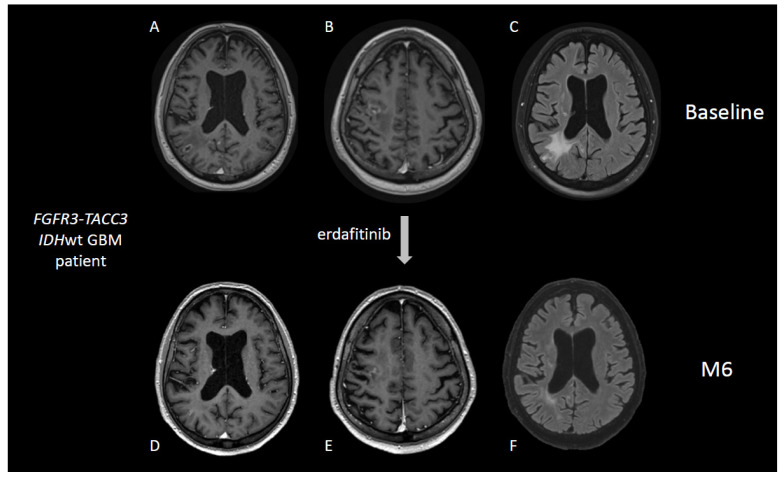
57-year-old patient with an IDH wild-type, FGFR3-TACC3 fusion positive GBM treated at recurrence with the FGFR inhibitor erdafitinib. Brain MRI imaging at baseline (**Panels A**–**C**) and after 6 months of therapy (**Panels D**–**F**).

**Figure 3 cancers-14-01124-f003:**
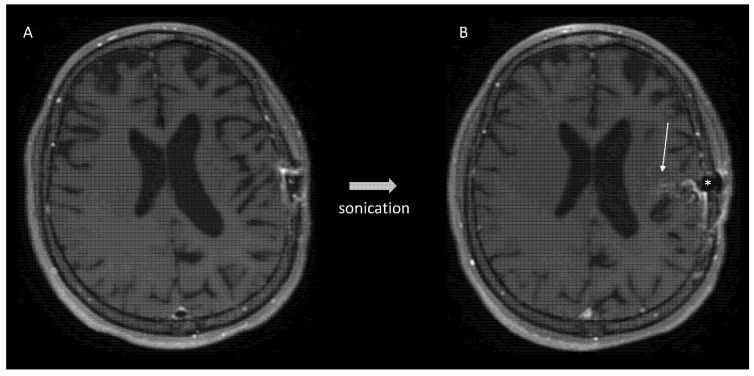
Adult patient with recurrent glioblastoma before (**Panel A**) and after (**Panel B**) sonication using the Sonocloud device (*star*) implanted in the skull. In (**Panel B**), contrast enhancement (*arrow*) indicates ultrasound mediated blood–brain barrier opening. Case already published in ref. [169].

**Table 1 cancers-14-01124-t001:** Trials of reference for conventional anti-tumor treatments in glioma patients. ECOG = Eastern Cooperative Oncology Group Performance Status Scale, Gy = Gray, KPS = Karnofsky Performance Status, NCT ID = National Clinical Trials identifier, PCV = procarbazine, lomustine, and vincristine polychemotherapy, RT = radiotherapy, TMZ = temozolomide, yo = year old.

Histo-Molecular Subgroup	Clinical Features	Therapeutical Intervention	NCT ID	Reference
Glioblastoma	KPS ≥ 70 and age ≤ 65 yo	Concomitant RT (60 Gy) + TMZ followed by adjuvant TMZ	NCT00006353	Stupp et al. NEJM 2005 [21]
	Age > 65 yo	Short-course concomitant RT (40 Gy) + TMZ followed by adjuvant TMZ	NCT00482677	Perry et al. NEJM 2017 [22]
	Age ≥ 70; KPS ≤ 70	TMZ	NCT01242566	Pérez-Larraya et al. JCO 2011 [23]
	KPS ≥ 60 and ≥65 yo; pMGMT methylated	TMZ	NCT01502241	Wick et al. Lancet Oncol 2012 [24]
	KPS ≥ 60 and ≥65 yo; pMGMT non-methylated	RT (60 Gy)	NCT01502241	Wick et al. Lancet Oncol 2012 [24]
Grade 3 oligodendroglioma, *IDH* mutated and 1p19q co-deleted	KPS ≥ 60	PCV followed by RT (59.4 Gy)	NCT00002569	Cairncross et al. JCO 2013 [25]
	ECOG ≤ 2	RT (59.4 Gy) followed by PCV	NCT00002840	Van den Bent et al. JCO 2013 [26]
Grade 3 astrocytoma, *IDH* mutated	KPS ≥ 60	PCV followed by RT (59.4 Gy)	NCT00002569	Cairncross et al. JCO 2013 [25]
	ECOG ≤ 2	RT (59.4 Gy) followed by adjuvant TMZ	NCT00626990	Van den Bent et al. The Lancet 2017 [27]
Grade 2 astrocytoma, *IDH* mutated	KPS ≥ 60; subtotal resection or age ≥ 40 yo	RT (54 Gy) followed by PCV	NCT00003375	Buckner et al. NEJM 2016 [28]

**Table 2 cancers-14-01124-t002:** Innovating strategies and targeted therapies: completed and recruiting trials. CNS = central nervous system, ECOG = Eastern Cooperative Oncology Group Performance Status Scale, KPS = Karnofsky Performance Status, LIPU: low intensity pulsed ultrasound, NCT ID = National Clinical Trials identifier, RT = radiotherapy, TMZ = temozolomide, yo = year old.

Histo-Molecular Subgroup and Disease Stage	Clinical Features	Therapeutical Intervention	NCT ID, Status	Reference
Recurrent glioblastoma	ECOG 0–1	Regorafenib	NCT02926222, completed	Lombardi et al. Lancet Oncology 2019 [39]
Recurrent glioblastoma and grade 3 astrocytoma	ECOG 0–1	Regorafenib plus nivolumab	NCT04704154, recruiting	
Newly diagnosed and recurrent glioblastoma	KPS ≥ 60	TMZ, lomustine, paxalisib, or VAL-083 (Bayesian response adaptive randomization)	NCT03970447, recruiting	
Recurrent *BRAF* V600E-mutant glioma	ECOG ≤ 2	Vemurafenib	NCT01524978, completed	Kaley et al. JCO 2018 [44]
Recurrent *BRAF* V600E-mutant glioma	ECOG ≤ 2	Dabrafenib and trametinib	NCT02034110, completed	Wen et al. Lancet Oncol 2022 [45]
FGFR3-TACC3+ recurrent glioblastoma	ECOG ≤ 2	AZD4547	NCT02824133, completed	
FGFR3-TACC3+ or FGFR1 mutant recurrent gliomas	ECOG 0–1	TAS120	NCT02052778, active (not recruiting)	
Recurrent solid tumors in CNS harboring NTRK Fusions	ECOG ≤ 3	Larotrectinib	NCT02576431, recruiting	
*IDH1* mutated advanced glioma	ECOG 0–1	Ivosidenib	NCT02073994, active (not recruiting)	Mellinghoff et al. JCO 2020 [46]
*IDH1* or *IDH2* mutated recurrent or progressive glioma	ECOG ≤ 2	Vorasidenib	NCT02481154, active (not recruiting)	Mellinghoff et al. Clin Cancer Res. 2021 [47]
Residual or recurrent *IDH* mutated grade 2 glioma	KPS ≥ 80	Vorasidenib	NCT04164901, recruiting	
Contrast enhancing *IDH1* mutated glioma	ECOG 0–1	Ivosidenib plus Nivolumab	NCT04056910, recruiting	
Recurrent *IDH* mutated grade 2 and 3 glioma	KPS > 50	Azacytidine	NCT03666559, recruiting	
**Immunotherapy**
Recurrent glioblastoma	KPS ≥ 60	Relatlimab with or without nivolumab	NCT02658981, recruiting	
Recurrent glioblastoma	KPS ≥ 60	INCMGA00012 and Epacadostat in Combination with RT and Bevacizumab	NCT03532295, recruiting	
Newly diagnosed glioblastoma	KPS ≥ 70	Nivolumab, BMS-986205, and RT with or without Temozolomide	NCT04047706, recruiting	
Recurrent glioblastoma with tumor mutational burden ≥ 10	ECOG ≤ 2	Ipilimumab and Nivolumab	NCT04145115, recruiting	
Recurrent glioblastoma with MMP2 expression	KPS ≥ 60	Chlorotoxin-CAR T-lymphocytes	NCT04214392, recruiting	
**Gene therapy and virotherapy**
Recurrent glioblastoma	Age ≤ 75 yo and KPS ≥ 70	Ad-RTS-hIL-12 plus veledimex and cemiplimab	NCT04006119, completed	
Surgically accessible recurrent glioblastoma	KPS ≥ 70	VB-111 neoadjuvant and adjuvant versus adjuvant only versus bevacizumab	NCT04406272, ongoing	
Recurrent glioblastoma	KPS ≥ 70	DNX-2401 plus pembrolizumab	NCT02798406, completed	
Newly diagnosed grade 3 and 4 glioma	KPS ≥ 70	Ad-TK + Ad-Flt3L combination therapy	NCT01811992, completed	
**Blood–brain barrier disruption**
Recurrent glioblastoma	KPS ≥ 70	LIPU and carboplatine	NCT02253212, completed	Idbaih et al. Clin Cancer Res. 2019 [48]
Recurrent glioblastoma	KPS ≥ 70	LIPU and carboplatine	NCT03744026, completed	
Newly diagnosed IDH wildtype glioblastoma	Age ≤ 70 yo and KPS ≥ 70	LIPU plus concurrent chemoradiation and adjuvant temozolomide	NCT04614493, recruiting

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
