# Peer review of "Innovating Strategies and Tailored Approaches in Neuro-Oncology"

_cancers, 2022, doi:10.3390/cancers14051124_

Round 1
Reviewer 1 Report
The paper “Innovating Strategies and Tailored Approaches in Neuro-Oncology by Alberto Picca et al. deals with a very critical issue. Authors aimed at discussing the most relevant diagnostic molecular markers of diffuse gliomas and their role in improving the classification of CNS tumours, the main prognostic markers, and eventually the “theranostic” markers with their corresponding targeted therapies currently in study for glioma patients. Authors also claim to present innovating promising strategies from recent clinical trials, but some of the most recent and promising approaches to glioma treaments are missing. This is a critical point especially if we consider that authors themselves report in the conclusion section:
- The efficacy of conventional antineoplastic treatments remains very limited in patients with malignant gliomas
- Several new approaches to increase their efficacy are currently under investigation.
So as Major revisions Authors are requested:
- to deepen the study of new strategies to manage glioma therapies;
- to increase he number of references to be cited.
As item 1. is concerned:
Gene therapy and Viro-immunotherapy
Gene therapy is a therapeutic approach that consists in utilizing genetic elements in order to treat or prevent disease. Whole genes, regulatory elements or oligonucleotides may be delivered to the target cells in glioma patients either by mechanical methods or using delivery vehicles. In order to achieve high therapeutic efficacy, gene therapy vectors must be chosen with caution, taking into consideration therapeutic transgene expression levels, distribution of gene expression within the TME, immunogenicity and biosafety. Gene therapy viral and nonviral vectors have shown efficacy in many pre-clinical studies since their first development in the 90s, but their clinical implementation still presents many challenges that must be overcome. One of the advantages of gene therapy is that its local administration may overcome the challenges posed by the BBB for systemic delivery approaches. Virotherapy is also an attractive therapeutic approach for glioma; it entails the use of genetically engineered viruses, which are no longer virulent and thus, cannot cause disease, but have the capacity of replicating within tumor cells, causing tumor cell death and release of oncolytic viral particles which can continue to infect and kill neighboring tumor cells.
Viral Vectors for Gene Therapy
Non-viral Vectors for Gene Therapy to overcome the BBB. Non-viral vectors are emerging as attractive platforms for gene therapy approaches for GBM such as liposomes and nanoparticles. Moreover, as drug penetration in the brain is an issue for GBM treatment, different ways of administering these agents are being assessed and, so far, intracranial delivery, though invasive, has demonstrated to be the most efficient in several approaches. Nanoparticles have emerged as a new and safe method for the delivery of agents targeting brain tumors and preclinical results are encouraging. It would be interesting to test the efficacy of these particles for the delivery of immune-stimulatory agents in the clinical setting. Finally, there is an urgent need for increased translational research and novel clinical trials to determine the potential efficacy of these novel therapies in glioma patients.
Combined therapies
In an effort to overcome the shortcomings of monotherapies, combination therapies have been developed. An example the human Phase-I dose escalation trial (NCT01811992) using a combination of two adenoviral vectors expressing HSV1-tk and Flt3L for the treatment of newly diagnosed, resectable malignant gliomas reported biological activity as evidenced by increased frequencies of DCs, CD4 and CD8 T cells within the TME It has recently become apparent that there is a need for combinatorial treatments in order to elicit higher therapeutic efficacy and better outcomes in the clinical arena. Combinatorial immune-gene therapies offer promising approaches for improving patient survival in GBM. Considering the numerous therapeutic approaches developed, the several possible targets, the improved current SOC and alternative dosing regimens and delivery routes, the number of potential combinations has increased exponentially. Several combinatorial approaches are today under clinical trials. In this respect, results from a Phase I clinical trial in which anti-PD-L1 was administered before and after GBM resection, demonstrated the importance of the selection of the starting point of the treatment.
Minor revisions
Authors are requested to present all data related to treatment protocols in a dedicated table instead of in the main text.
Author Response
Authors: We thank the reviewer for her/his commentaries and suggestions. We reply point by point in the following text.
Reviewer: So as Major revisions Authors are requested:
to deepen the study of new strategies to manage glioma therapies;
Response: we thank the reviewer for her/his suggestions. We further developed the manuscript with new dedicated paragraphs on CAR T cells therapy, gene therapy and virotherapy approaches including references to related clinical trials.
to increase the number of references to be cited.
Response: we increased the number of references according to the reviewer’s request.
Authors are requested to present all data related to treatment protocols in a dedicated table instead of in the main text.
Response: we thank the reviewer for her/his suggestion. We added two tables listing 1) trials of reference for the management of glioma patients with conventional anti-tumor treatments 2) completed or recruiting clinical trials testing innovating strategies and targeted therapies. Major histo-molecular and clinical features of target populations are detailed together with the reference when available and the NCT registration numbers.
We hope that the revised manuscript is now suitable for publication.
Reviewer 2 Report
In the manuscript titled “Innovating Strategies and Tailored Approaches in Neuro-Oncology” Picca A et. al. review the diagnostic molecular markers and their role in the classification of diffuse gliomas, which are the most common primary central nervous system (CNS) tumors. Further, the authors discuss the targeted therapies currently in use or clinical trials in glioma patients. Finally, the authors list innovative therapeutic strategies under study to treat glioma.
The review summarizes current knowledge on the classification and treatment of gliomas. However, all of the information is in text form; the authors should consider using tables to make the information easily accessible to the general audience and increase the reach of the manuscript. The section on innovative therapeutic strategies is not comprehensive. Many strategies such as cell therapy and gene therapy-based approaches are not discussed. The authors should consider including these strategies for completeness. A few typos/grammatical issues are present in the manuscript for example:
“Targeted treatments, along with improved immunotherapeutic schedules and physical strategies to improve drug delivery to the nervous system, are currently under extensive investigation and bring hope for mor effective therapies in these diseases with currently often a dismal outcome.”
“Encouraging data derive from the use of regorafenib (Figure 1A-D), an oral multi-kinase inhibitor targeting VEGFR1-3, PDGFR, TIE2, FGFR, RAF-1, KIT, RET and BRAF, evaluated for recurrent GBM patients in the randomized phase II trial REGOMA”.
Overall, the review is well written, sufficiently cited, and will be of interest to the scientific community studying CNS tumors. This review will be suitable for publication in Cancers with some modifications.
Author Response
Authors: We thank the reviewer for her/his commentaries. We reply point by point in the following text.
Reviewer: The review summarizes current knowledge on the classification and treatment of gliomas. However, all of the information is in text form; the authors should consider using tables to make the information easily accessible to the general audience and increase the reach of the manuscript.
Response: we thank the reviewer. As replied to Reviewer 1, we added two tables detailing references or active clinical trials as requested.
Reviewer: The section on innovative therapeutic strategies is not comprehensive. Many strategies such as cell therapy and gene therapy-based approaches are not discussed. The authors should consider including these strategies for completeness. A few typos/grammatical issues are present in the manuscript
Response: We thank the reviewer for these suggestions. We verified again the text for English content and made corrections as requested.
We hope that the revised manuscript is now suitable for publication.
Round 2
Reviewer 1 Report
Authors have made effort to comply the request of major revisions. The paper can now be accepted for publication in Cancers.
Minor revision: Please check the reference 172 that is reported in Reference Section but it is not cited in the text.